# Resilience as a Mediator of the Association between Spirituality and Self-Management among Older People with Chronic Obstructive Pulmonary Disease

**DOI:** 10.3390/healthcare9121631

**Published:** 2021-11-25

**Authors:** Zhongyi Chen, Yuyu Jiang, Mengjie Chen, Nuerdawulieti Baiyila, Jiang Nan

**Affiliations:** Research Office of Chronic Disease Management and Rehabilitation, Wuxi School of Medicine, Jiangnan University, No.1800 Lihu Avenue, Wuxi 214122, China; 6192806002@stu.jiangnan.edu.cn (Z.C.); ccchenmengjie@163.com (M.C.); 18861877297@163.com (N.B.); 6212807061@stu.jiangnan.edu.cn (J.N.)

**Keywords:** chronic obstructive pulmonary disease, spirituality, resilience, self-management, mediating role

## Abstract

This study examined the mediating effect of resilience in the relationship between spirituality and self-management among older people with chronic obstructive pulmonary disease (COPD). The participants were 151 older people with COPD in four general hospitals in Jiangsu Province, China. Data were collected from September 2020 to May 2021 using a questionnaire developed by the investigator, the Function Assessment of Chronic Illness Therapy-Spiritual Scale (FACIT-SP-12), 10-item Connor-Davidson Resilience Scale (CD-RISC-10), and COPD Self-Management Scale (CSMS). One-way ANOVA and *t*-test were used to compare the level of self-management in patients with different sociodemographic and clinical characteristics. Partial correlation analysis was used to explore the correlation between spirituality, resilience, and self-management. Hierarchical multiple regression analyses were performed to examine the contribution of spirituality and resilience to the prediction of self-management. A bootstrapping test was implemented using the SPSS PROCESS macro to test the statistical significance of the mediating effect. There was a pairwise positive correlation between spirituality, resilience, and self-management. Resilience mediated the relationship between spirituality and self-management. These findings suggested that resilience interventions could be incorporated into future COPD self-management interventions to better improve self-management and health outcomes. Moreover, resilience should be an important component of healthy aging initiatives.

## 1. Introduction

Chronic obstructive pulmonary disease (COPD) is a chronic respiratory disease characterized by persistent respiratory symptoms and irreversible airflow limitation [1]. COPD currently ranks as the third leading cause of death and the fifth leading economic burden of disease worldwide [2,3,4].

Once affected, patients will need lifelong treatment and disease management. Even with optimal treatment, the progression of the disease can only be slowed, and many COPD patients will inevitably experience a variety of discomforts (e.g., dyspnea, cough, sputum, fatigue, depression, etc.) and acute exacerbations [5,6]. Therefore, in addition to maintaining good compliance with treatment, COPD patients must learn how to cope with symptoms and emotions, adapt their lifestyles to the limitations of the disease, and know how to prevent, recognize, and manage exacerbations [1,7]. Therefore, COPD is a chronic disease that relies heavily on patient self-management. Self-management, a key element in the management of chronic diseases [8,9], is defined as the acquisition by patients of the skills necessary to manage and cope with their illness in their daily lives and to adopt and adhere to behaviors that improve their health [5]. Effective self-management can reduce symptoms of breathlessness [10], improve quality of life [7,11], and reduce hospital admissions [12]. Previous studies have shown that the self-management level of COPD patients is low [13,14]. Relevant reviews and investigative studies have found that the level of self-management in COPD patients is strongly associated with demographic factors (e.g., age), physical factors (e.g., dyspnea), psychological factors (e.g., depression and resilience), and spiritual factors (e.g., meaning of life and beliefs) [8,13,15]. Spirituality belongs to an inner force. It is a way for individuals to find hope, meaning, and purpose [16], and plays an important role in the face of illness or crisis, helping patients to come to terms with their illness and maintain good compliance with treatment [17,18]. The role of spirituality in the management of COPD chronic illness has received attention in the last decade [16]. Spiritual support is also proposed in the Global Initiative for Chronic Obstructive Lung Disease as an important component of palliative care [1], but the number of relevant studies remains limited. The impact of spirituality on COPD self-management needs to be further explored.

Spiritual health, as the fourth dimension of health [19], is often used to reflect the state of health at the spiritual level [20]. John et al. [21] define it as an individual’s exploration of the meaning and purpose of life in dealing with relationships with self, others, the environment and the supernatural, and the search for self-integration, leading to a state of inner peace and comfort. People with high spiritual health are able to have a high sense of self-efficacy [22,23], cope positively with illness, and adopt health-related behaviors and lifestyles that contribute to self-management [16,24,25,26]. Helvaci et al. [27] showed a significant positive correlation between spiritual health and medication adherence in COPD patients. Mendes et al. [28] found that COPD patients with high levels of spiritual health showed less anxiety and depression, as well as better emotional functioning and disease control, suggesting better emotional management. Both medication management and emotional management are part of COPD self-management [29,30]. Therefore, there is a correlation between spirituality and self-management in COPD patients. Previous studies have also shown that spirituality is closely related to the self-management of chronic diseases [31,32,33,34]. However, current research exploring the relationship between spirituality and chronic disease self-management is limited to qualitative studies, with few investigative studies [31,32,33,34]. There are also no studies that correlate spirituality with self-management level in COPD patients.

In addition, the influence of spirituality on self-management in COPD patients may not necessarily be direct but may act indirectly through resilience. The American Psychological Association defines resilience as the ability of an individual to bounce back from a difficult experience by adapting well in the face of adversity, trauma, tragedy, threat, or other significant stress [35]. For COPD patients, the main stressors they face are the direct and indirect effects of disease symptoms, management, and treatment, including painful dyspnea symptoms and the adverse emotions they cause, such as anxiety, depression, and helplessness, difficulty coping with or adapting to disease-induced activity limitations (physical and social recreational activities), reduced self-care and family dependence, and the stress of progressive disease exacerbation and recurrent episodes [36,37,38]. Luckett et al. [39] concluded that resilience strategies help to promote and maintain self-management behaviors, such as dyspnea self-management behaviors, in patients with COPD. Disler et al. [8] found that many positive factors (e.g., optimism, hope, sense of control) were found to influence the self-management of people with COPD through resilience. The positive correlation between resilience and self-management was confirmed in diabetic and hemodialysis patients. [40,41]. In addition, existing studies have shown that spirituality was positively correlated with resilience and was a significant predictor of resilience [42,43,44]. A review by Disler et al. [8] found that religion and beliefs that fall under the category of spirituality influence self-management in COPD patients through resilience, suggested that resilience plays a mediating role in the relationship between spirituality and self-management, but there is a lack of quantitative research evidence for this result.

With the aging of the world population, the management of chronic diseases in older populations is receiving increasing attention, and the prevalence of COPD in the older population is at a high level [1]. Therefore, this study aims to elucidate the relationship and intrinsic connection between spirituality, resilience, and self-management ability of older COPD patients from the perspective of positive psychology, enrich the self-management intervention strategies, and provide new perspectives for self-management intervention and healthy aging actions.

Based on the above, we hypothesized that: (a) positive associations between spirituality and self-management would be found, (b) positive associations between spirituality and resilience would be found, (c) positive associations between resilience and self-management would be found, and (d) the associations between spirituality and self-management would be mediated by resilience.

## 2. Materials and Methods

### 2.1. Participants and Procedure

This study was a cross-sectional study conducted from September 2020 to May 2021 and was reviewed and approved by the Medical Ethics Committee of Jiangnan University (JNU20190318IRB61). Older COPD patients in the respiratory medicine wards of 4 general hospitals in Jiangsu Province, China, were recruited as study subjects. COPD patients who met the diagnostic criteria in the Guidelines for the diagnosis and management of chronic obstructive pulmonary disease (revised 2021) [1] and were ≥60 years old, had no communication impairment, could respond correctly, could complete the questionnaire independently or with the help of the investigator, and volunteered to participate in this study were included. Patients with severe mental or cognitive impairment, hearing or speech dysfunction, or with other serious medical conditions were excluded. Of the 185 patients who were interested in participating and screened for eligibility, 15 were excluded.

Prior to the survey, the investigator briefed the patient on the purpose and process of the study and obtained informed consent from the patient. After informed consent was obtained, patients were guided by the investigator to complete the questionnaire. The questionnaire covered 4 main sections: patient sociodemographic and clinical information, spirituality, resilience, and COPD self-management. The entire questionnaire took between 20–30 min to complete.

We used G. Power 3.1.9.7 to calculate the appropriate sample size. A sample size of at least 144 patients was needed for multiple regression analysis based on an effect size of f^2^ = 0.15, α of 0.05, a statistical power of 0.85 and 13 independent variables. As a result, a total of 170 patients were selected for the survey, of whom 151 participated in the survey, making the participation rate 88.8%.

### 2.2. Measures and Variables

Sociodemographic and clinical data were collected using a questionnaire developed by the investigator, including gender, age, marital status, education level, monthly income, smoking status, religion, number of hospitalizations in the last 1-year, Chronic Obstructive Lung Disease (GOLD) stage, degree of dyspnea, and depression. The modified Medical Research Council Dyspnea Scale (mMRC) was used to assess the degree of dyspnea in COPD patients, and dyspnea was classified into 5 grades from 0–4, with higher grades indicating more severe dyspnea [45]. The 15-item Geriatric Depression Scale (GDS-15) was used to discriminate between depressive and non-depressive states. It contained 15 items, in which patients were asked to respond by answering “yes” or “no” about how they feel. Items 1, 5, 7, 11, and 13 were scored 1 point for “no” answers and 1 point for “yes” answers to the other items. The normative score ranges for the GDS-15 were “normal” (0–4), “mild” (5–9), and “moderate to severe” (10–15) [46].

The Function Assessment of Chronic Illness Therapy-Spiritual Scale (FACIT-SP-12) was used to assess the spirituality of COPD patients [27,28,47]. The scale contained 3 dimensions of peace, meaning, and faith, with 12 entries. Each entry was scored on a Likert 5-point scale, with a score of 0 indicating no at all and 4 indicating very much. The total scale score was 0–48, with higher scores indicating higher spiritual health. As this study was conducted in China, the Chinese version of the FACIT-SP-12, which was Chinese translated by Liu et al. [48] was used. The Cronbach’s alpha coefficient for the scale was 0.831 and the content validity was 0.90. Total scores <24, 24 to 35, and ≥36 represented low, medium, and high levels of spirituality, respectively.

The 10-item Connor-Davidson Resilience Scale (CD-RISC-10) was used to assess patients’ resilience [49]. The scale contained 10 items, each of which was scored on a Likert 5-point scale, with 0 indicating never and 4 indicating always, and higher scores indicating higher levels of resilience. This study used the Chinese version of the CD-RISC-10, which was Chinese translated by Zhang et al. [50]. The scale had good reliability and validity, with a Cronbach’s α of 0.737 and a retest reliability of 0.974.

The COPD Self-Management Scale (CSMS), developed by Zhang et al. [30], was used to assess patients’ self-management level. The scale contained 5 dimensions of symptom management, daily life management, emotion management, information management, and self-efficacy, with a total of 51 entries. Each entry was rated on a 5-point Likert scale, with 1 indicating none and 5 indicating always, with a total score of 51–255, with higher scores indicating higher levels of self-management. The CSMS showed good reliability and validity in the validation study. The test-retest correlation coefficient and Cronbach alpha coefficient of the CSMS were 0.87 and 0.92, respectively. The content validity index of the CSMS was 0.90 [30]. The mean and standard deviation (SD) of the scores of all patients were obtained. Patients were considered to have a high level of self-management when their CSMS scores were greater than the mean plus 1 SD; those with scores less than the mean minus 1 SD were considered to have a low level of self-management; and those with scores greater than or equal to mean minus 1 SD and less than or equal to mean plus 1 SD were classified as having a medium level of self-management [13,51].

### 2.3. Statistical Analysis

Statistical analysis was performed using SPSS 24.0 statistical software. Sociodemographic and clinical data, spirituality scores, resilience scores, and COPD self-management scores were analyzed using descriptive statistics. Before data analysis, the Shapiro–Wilk test was used to check the normality of numeric variables. One-way ANOVA or *t*-test was used to compare the level of self-management in patients with different sociodemographic and clinical characteristics. Using partial correlation analysis, the correlation between spirituality, resilience, and self-management were analyzed after controlling for the effect of sociodemographic and clinical variables. Hierarchical multiple regression (HMR) analyses were performed to examine the contribution of spirituality and resilience to the prediction of self-management. Finally, we performed a bias-corrected bootstrapping analysis (with 5000 resamples) using the SPSS PROCESS Macro Model 4 to verify the mediating effects of resilience [52,53]. This approach was a suitable option for research where the variables are all directly measured variables [54]. The mediation analysis by SPSS PROCESS macro was based on regression-based path analysis. Path coefficients (a, b, c, and c’) in the mediation model were obtained after analysis. The path analysis can be divided into 3 steps. In the first step (Model 1), the total effect of spirituality (predictor variable) on self-management (outcome variable) was estimated (c path). In the second step (Model 2), the relationship between spirituality (predictor variable) and resilience (mediator variable) was estimated (a path). Finally (Model 3), the direct effect of spirituality (predictor variable) on self-management (outcome variable) (c’ path), and the relationship between self-management (outcome variable) and resilience (mediator variable) while the spirituality was controlled (b path) was estimated (as shown in Figure 1). [55]. The mediation effect (c − c’ = ab, ab was also known as “indirect effect”) is indicated by a statistically significant difference between c and c’ [56]. The mediating effect was considered statistically significant if the 95% bootstrap confidence interval of indirect effect did not contain zero [57]. All statistical tests were performed at a 0.05 level of significance.

## 3. Results

### 3.1. Participants’ Sociodemographic and Clinical Characteristics

Descriptive data on patient sociodemographic and clinical characteristics are shown in Table 1. A total of 84.1% of the included patients were male, and 15.9% were female. The mean age of the older patients with chronic obstructive pulmonary disease was (74.36 ± 7.34) years, with an age range of 60–94 years; 79.5% of the patients had a surviving spouse; 39.7% of the patients had an educational level of primary school or below, 51.0% had attended junior-senior high school, and only 9.3% had attended college or higher; monthly income (Yuan) with <3000, 3000–5000 and >5000 were 40.4%, 29.1%, and 30.5%, respectively. More than half of the patients (53.0%) were ex-smokers, and 24.5% had never smoked; only 9.9% of the patients had religious beliefs; The proportions of hospitalizations in the last year of 0–1, 2–3, and >3 were 28.5%, 60.9%, and 10.6%, respectively. The proportions of patients with GOLD stage I, II, III, and IV were 1.3%, 18.5%, 41.1%, and 39.1%, respectively. A total of 13.9% of patients had mMRC less than grade 2, and 86.1% had grade 2 or above; 19.9% of patients had different levels of depression. There was no difference in the level of self-management among COPD patients by gender, age, marital status, smoking status, religion, and the number of hospitalizations. However, there were differences in self-management levels by education level, monthly income, GOLD stage, mMRC grade, and depression. A comparison of the levels of self-management among COPD patients with different sociodemographic and clinical characteristics is shown in Table 1.

### 3.2. Descriptive Characteristics of Target Variables and Relationships among Them

Descriptive characteristics of target variables are shown in Table 2. The mean score of FACIT-SP-12 was 26.77 ± 7.47, at a moderate level; the mean score of CD-RISC-10 was 23.96 ± 6.21; and the mean score of CSMS was 151.38 ± 21.21. Based on the mean plus or minus one standard deviation of the self-management scores, those with scores greater than one mean plus one standard deviation (>172) were considered to have a high level of self-management; those with scores less than the mean minus one standard deviation (<131) were considered to have a low level of self-management; those with scores between 131 and 172 were considered to have a medium level of self-management. Thus, 16.5%, 68.9%, and 14.6% of the respondents were considered to have low, moderate, and high levels of self-management, respectively. The standard scores for each subscale of self-management were, from high to low, daily life management, emotion management, self-efficacy, symptom management, and information management. The Shapiro–Wilk tests were performed for all target variables and did not show evidence of non-normality.

The relationships between target variables are shown in Table 3. Significant correlation was found between spirituality and resilience (*r* = 0.437, *p* < 0.01). All dimensions except the information management dimension of self-management were positively correlated with spirituality. All dimensions of self-management were found to be positively associated with resilience. 

R^2^ and adjusted R^2^ were included in Table 4 to better reflect the size-effects of the correlations of self-management with spirituality and resilience.

As is indicated in Table 4, spirituality was significantly positively correlated with self-management, accounting for 12.9% of the variance. Resilience was also significantly positively correlated with self-management, explaining an additional 8.9% of the variance. The results indicate the potential effect of spirituality on self-management might be partially mediated by resilience. The regression coefficient (β) for the association between spirituality and self-management was reduced from 0.420 to 0.250 when resilience was added to the model.

### 3.3. Mediating Effects of Resilience on the Relationship between Spirituality and Self-Management 

We used Model 4 in SPSS PROCESS macro to test the mediating effect of resilience on the association between spirituality and self-management. Because self-management level in COPD varies by education level, monthly income, GOLD stage, mMRC grade, and depression, the significance of the direct, indirect, and total effects in the mediation model was identified after controlling for these five variables (as shown in Table 5).

In Model 1, controlling for education level, monthly income, GOLD stage, mMRC grade, and depression, spirituality was significantly associated with self-management (β = 0.4198, *p* < 0.0001). In Model 2, controlling for education level, monthly income, GOLD stage, mMRC grade, and depression, spirituality was significantly associated with resilience (β = 0.4491, *p* < 0.0001). In Model 3, controlling for education level, monthly income, GOLD stage, mMRC grade, and depression, both spirituality and resilience were included in the mediation model and showed a significant relationship with self-management. Simultaneously, the standardized regression coefficient (β) for spirituality decreased from 0.4198 to 0.2502 (as shown in Table 5).

Moreover, the results of the non-parametric bootstrapping method confirmed the significance of the indirect effect of spirituality on self-management through resilience (95% bootstrap CI = 0.2243, 0.7671). A bootstrapped 95% confidence interval (CI) confirmed that the indirect effect of spirituality had an impact of 0.4813 that was produced by resilience as a mediator on self-management. The indirect effect of resilience accounted for 40.39% of the total variance in self-management influenced by spirituality (as shown in Table 6). These findings corroborate our hypothesis that resilience may play a mediator role in the association between spirituality and self-management. Figure 2 illustrates the mediation model, along with standardized path coefficients (a, b, c, c’).

## 4. Discussion

This study found that older COPD patients had low levels of self-management. Spirituality and resilience of older COPD patients were positively associated with self-management, and resilience mediated the association between spirituality and self-management.

The results of this study showed that the vast majority (85.4%) of older COPD patients had a moderate or low level of self-management. This is similar to the results of previous related studies [13,14]. This study found that older COPD patients with higher levels of education tended to have higher levels of self-management. The reason for this difference may be partly due to the higher level of health literacy in accessing, understanding, and applying disease-related information and skills in patients with higher education levels [58], which is associated with a higher self-management level [14]. On the other hand, patients with higher education levels tend to have higher self-efficacy and thus better treatment adherence and confidence to implement effective self-management [59]. In addition, this study found that older COPD patients with high GOLD stage and severe dyspnea exhibited lower levels of self-management. Dyspnea is the most common symptom of chronic obstructive pulmonary disease and can recur due to a variety of stimuli such as physical activity, ambient temperature, health status, and emotional state, and is often difficult for patients to predict and cope with [8,60]. Patients with severe dyspnea often experience discomfort during daily activities and even at rest and are associated with restricted and reduced activity, reduced quality of life levels, fear, anxiety, and depression [61,62,63], which may lead to reduced motivation to undertake disease management.

Spirituality as a coping mechanism mobilizes the patient’s inner and social resources to manage various stressful events in life and contributes to self-management level [16,64]. People with high levels of spiritual health believe that life has meaning, value, and purpose and have faith [65]. This study is the first to demonstrate a direct positive effect of spiritual health on self-management of COPD patients from a quantitative perspective, and previous related studies support the results of this study to some extent. Sheridan and Seamark et al. [66,67] found that when COPD patients perceive the value and meaning of life, they are more likely to adopt positive health self-management behaviors. Conversely, when COPD patients experience an increased burden of painful disease and a decrease in enjoyment of life and social activities, patients lose their purpose in life and question the meaning of life, ultimately leading to a significant decrease in patient motivation to self-manage [66,68]. Religious beliefs fall under the category of spiritual health, and people with religious beliefs are more likely to develop self-management behaviors, such as eating healthy and quitting smoking [16]. The positive impact of spirituality on self-management has also been demonstrated in patients with type 2 diabetes, chronic kidney disease, and stroke [33,64,69]. Studies by Samuel-Hodge and Baig et al. [70,71] have both found greater benefits of self-management programs that incorporate spiritual interventions to enhance self-management in people with diabetes. The results of this study may provide supporting evidence for spiritual interventions in the chronic management of COPD patients.

Researchers believe that spirituality belongs to one of the dimensions of resilience and is a potential driver of resilience and is closely related to it [72]. The present study found a significant positive correlation between spiritual health and resilience, which is consistent with results in other diseases (depression, spinal cord injury, schizophrenia, and bipolar disorder) [42,44,73]. In addition, spirituality is often considered an important resilience resource and protective factor that can increase resilience through its impact on relationships, life values, personal meaning, and better coping with stress [74,75,76]. Jones’ study found that spiritual health, a significant predictor of resilience, explained more than half (51%) of the variance in resilience [42]. There is already research that includes elements that contribute to spiritual health-related content (e.g., finding positive meaning) as one of the resilience intervention strategies [77]. However, there are no studies on resilience interventions for patients with COPD. Both previous studies and the present study found that spiritual health in patients with COPD was at an intermediate level [27,78,79]. One study found that spiritual health scores in patients with COPD were similar to those of lung cancer patients and could be further improved [79]. Therefore, in future interventions for resilience in COPD, we can focus on enhancing resilience by improving patients’ spiritual health, and more effective spiritual intervention strategies (e.g., meaning therapy, life review therapy, etc.) can also be used as part of resilience interventions.

With the discovery of the benefits of enhancing one’s intrinsic positives in the management of chronic disease, scholars have begun to focus on elements in positive psychology [80]. Resilience is one of the most frequently cited positive psychology resources [81]. The concept of resilience is not yet uniformly understood by academics. Resilience is usually defined in chronic illness from three perspectives: (1) competence-based definition: the ability to adapt resiliently and flexibly to disease-related stress and stress reactions. (2) outcome definition: the ability to adapt well and demonstrate positive health outcomes despite the stress and threat of illness; and (3) process definition: the dynamic process of successful adaptation in the face of chronic illness [80]. The present study found a significant positive association between resilience and self-management, which can be explained behaviorally and content-wise. Behaviorally, patients with high levels of resilience tended to have more health-promoting behaviors (healthy diet, participation in exercise, good medication adherence, lifestyle, adherence to breathing exercises, etc.) [39,82,83,84], which are the components of self-management. Content-wise, the components of the resilience scale (e.g., self-efficacy, coping skills) have been shown to improve self-management in COPD patients [9,85]. The positive association between resilience and self-management has also been demonstrated in other chronic patients (hemodialysis patients, type 2 diabetes, etc.) [40,41]. Based on this, several researchers have explored the impact of resilience interventions on self-management of chronic disease and found positive results of improved self-management [77,86]. Steinhardt et al. [86] initially explored and validated the acceptability of a 4-week diabetes self-management program incorporating a resilience intervention and the effectiveness of improving self-management level, glycated hemoglobin, etc. Dubois et al. [77] developed a resilience-based self-management program and demonstrated its benefits in increasing self-management behaviors and physical activity, improving glycated hemoglobin and psychological well-being. However, most current interventions for COPD self-management are based on comprehensive self-management programs and have not yet involved resilience interventions [87]. Edward et al. recommended resilience interventions for healthcare professionals to improve patient self-management in chronic diseases such as chronic respiratory disease [88]. There is evidence that resilience in COPD patients is significantly and positively associated with quality of life levels [89]. Therefore, future studies should integrate resilience interventions into COPD self-management programs to better enhance patient’s self-management and improve health outcomes.

This study is the first to find that resilience in older COPD patients plays a mediating role in the relationship between spirituality and self-management from a quantitative perspective, and the same results have been mentioned in previous qualitative interviews with diabetic patients, such that spirituality can influence motivation and perseverance in self-management by key components of resilience (self-reliance, perseverance, peace) [90]. As countries focus on healthy aging, the social and personal determinants of health associated with healthy aging have become a concern for researchers. Resilience is a positive psychological resource inherent in the individual, which is mainly emphasized as an ability of the individual to recover from significant stress or distress and to adapt flexibly to changes in the external environment, which is dynamic and constantly growing and developing itself in the process of adapting to stress. In summary and in conjunction with the results of this study, the older people with a high level of resilience are able to cope with various challenges in the management of chronic diseases and become active participants in the healthcare delivery system and contribute to the achievement of healthy aging (good functioning, active participation in life roles, good quality of life levels). There is evidence that resilience is an independent predictor of healthy aging [82], and some researchers have argued that resilience should be an important component of healthy aging and a core component of self-management programs [82,91]. The results of the study further expand the understanding of resilience and its application to chronic disease management and healthy aging, providing ideas for future self-management interventions and healthy aging initiatives.

This study has certain limitations: First, the study design was a cross-sectional study and could not confirm the causal relationship between the variables. Future longitudinal studies should be conducted to verify the effects of spirituality and resilience on self-management and the mediating role of resilience. Second, as this study used convenience sampling rather than random sampling, there may be some deviations from the true results, and the survey was only conducted in Jiangsu province, thus the scope of application of the findings is limited. Future studies may consider random sampling and conducting surveys in multiple centers to confirm the results of this study.

## 5. Conclusions

Older people with COPD have low levels of self-management. There is a positive correlation between spirituality, resilience, and COPD self-management level, and spirituality indirectly influences self-management level through resilience. There is a need to integrate resilience interventions into COPD self-management programs in the future to better enhance patient self-management and improve health outcomes. In addition, resilience should be an important component of healthy aging initiatives.

## Figures and Tables

**Figure 1 healthcare-09-01631-f001:**
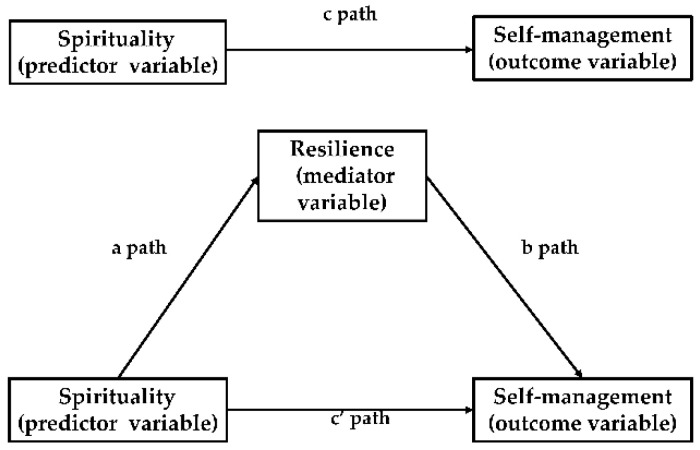
Diagram of paths in the mediation model.

**Figure 2 healthcare-09-01631-f002:**
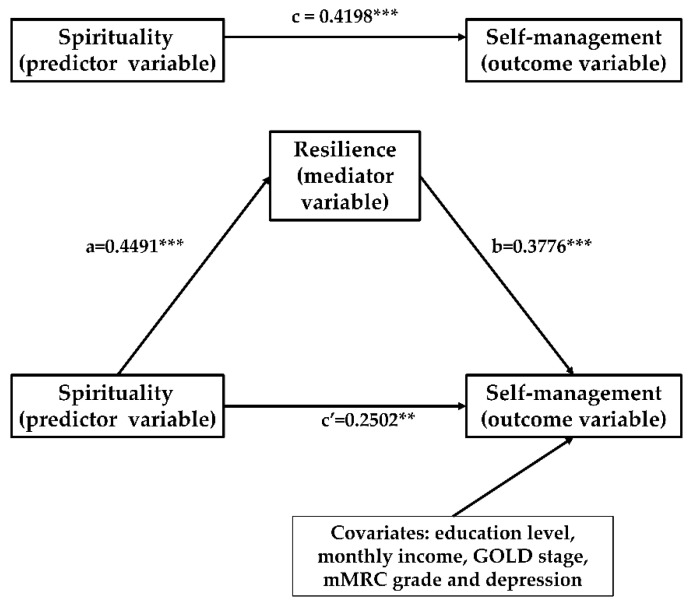
Proposed models that investigate mediated effects (** *p* < 0.01; *** *p* < 0.001).

**Table 1 healthcare-09-01631-t001:** Participant sociodemographic and clinical characteristics, differences of self-management level by characteristics.

Characteristics	Categories	n (%)	COPD Self-Management		
			M ± SD	*t*/F	*p*
Gender	Male	127 (84.1)	152.61 ± 20.88	1.638	0.104
	Female	24 (15.9)	144.92 ± 22.23		
Age (years)	60–69	35 (23.2) *	152.54 ± 16.52	0.749	0.475
	70–79	78 (51.7) *	152.64 ± 21.79		
	≥80	38 (25.2) *	147.74 ± 23.80		
Marital status	Spouses living	120 (79.5)	152.07 ± 21.38	0.777	0.438
	Others	31 (20.5)	148.74 ± 20.66		
Education level	Primary school and below	60 (39.7)	143.25 ± 17.48	8.991	<0.001
	Junior-senior high school	77 (51.0)	155.55 ± 21.83		
	College and above	14 (9.3)	163.36 ± 21.33		
Monthly income, Yuan	<3000	61 (40.4)	146.92 ± 21.14	3.627	0.029
	3000–5000	44 (29.1)	150.82 ± 20.12		
	>5000	46 (30.5)	157.85 ± 21.14		
Smoking status	Never smoked	37 (24.5)	147.54 ± 20.46	3.049	0.050
	Current smoker	34 (22.5)	146.29 ± 17.61		
	Ex-smoker	80 (53.0)	155.33 ± 22.36		
Are religious?	Yes	15 (9.9)	153.80 ± 14.86	0.464	0.644
	No	136 (90.1)	151.12 ± 21.82		
Number of hospitalizations in the last 1 year	0–1	43 (28.5)	153.95 ± 22.36	0.531	0.589
	2–3	92 (60.9)	150.73 ± 21.36		
	>3	16 (10.6)	148.25 ± 17.28		
GOLD stage	Stage I	2 (1.3)	198.00 ± 4.24	4.721	0.004
	Stage II	28 (18.5)	156.00 ± 21.76		
	Stage III	62 (41.1)	151.68 ± 19.95		
	Stage IV	59 (39.1)	147.31 ± 20.55		
mMRC	<2	21 (13.9)	168.57 ± 22.53	4.221	<0.001
	≥2	130 (86.1)	148.61 ± 19.71		
Depression	nomal	121 (80.1) *	153.21 ± 21.49	3.741	0.0260
	mild	26 (17.2) *	146.54 ± 17.91		
	moderate tosevere	4 (2.6) *	127.75 ± 16.30		

Note: M = mean; SD = standard deviation; *t* = values of *t*-test; F = values of ANOVA; COPD = chronic obstructive pulmonary disease; GOLD = Chronic Obstructive Lung Disease; mMRC = The modified Medical Research Council Dyspnea Scale; * because only one decimal is retained in the numerator of the percentages, the percentages of patients of different ages and the percentages of patients with different levels of depression do not add up to 100 percent, but the total number of patients is 151 in all cases.

**Table 2 healthcare-09-01631-t002:** Descriptive statistics of target variables.

Variables	Range	M ± SD	Standardized Score (M ± SD)
Spirituality	8–46	26.77 ± 7.47	2.23 ± 0.62
Resilience	9–39	23.96 ± 6.21	2.40 ± 0.62
Symptom management dimension	13–32	22.91 ± 3.71	2.86 ± 0.46
daily life management dimension	31–61	45.18 ± 6.58	3.23 ± 0.47
emotion management dimension	21–57	37.60 ± 6.42	3.13 ± 0.54
information management dimension	8–31	18.75 ± 4.76	2.34 ± 0.60
self-efficacy dimension	13–44	26.93 ± 5.94	2.99 ± 0.66
Self-management	90–210	151.38 ± 21.21	2.97 ± 0.42

Note: M = mean; SD = standard deviation.

**Table 3 healthcare-09-01631-t003:** Partial correlations between target variables after controlling potential confounders #.

Variables	Correlation Matrix							
	1	2	3	3-1	3-2	3-3	3-4	3-5
1. Spirituality	1							
2. Resilience	0.437 **	1						
3. Self-management	0.419 ***	0.497 ***	1					
3-1. Symptom management dimension	0.205 *	0.236 **	0.582 ***	1				
3-2. Daily life management dimension	0.324 ***	0.218 **	0.762 ***	0.307 **	1			
3-3. Emotion management dimension	0.439 ***	0.595 ***	0.786 ***	0.242 **	0.467 ***	1		
3-4. Information management dimension	0.158	0.297 ***	0.662 ***	0.435 ***	0.303 ***	0.377 **	1	
3-5. Self-efficacy dimension	0.360 ***	0.444 ***	0.853 ***	0.396 ***	0.587 ***	0.666 ***	0.424 ***	1

Note: # potential confounders: education level, monthly income, GOLD stage, and mMRC grade, and depression; * *p* < 0.05 (two-tailed); ** *p* < 0.01 (two-tailed); *** *p* < 0.001 (two-tailed).

**Table 4 healthcare-09-01631-t004:** The hierarchical multiple regression analyses of self-management.

	Self-Management		
	Model 1	Model 2	Model 3
Block 1: Sociodemographic and clinical characteristics			
Education level	0.283 ***	0.152 *	0.111
Monthly income	0.140	0.056	0.036
GOLD stage	−0.035	−0.006	−0.063
mMRC grade	−0.270 **	−0.180 *	−0.087
Depression	−0.220 **	−0.168 *	−0.124
Block 2: Spirituality	/	0.420 ***	0.250 **
Block 3: Resilience	/	/	0.378 ***
R^2^	0.264	0.393	0.483
Adjusted R^2^	0.239	0.368	0.457
∆ R^2^	0.264	0.129	0.089

Note: * *p* < 0.05 (two-tailed); ** *p* < 0.01 (two-tailed); *** *p* < 0.001 (two-tailed).

**Table 5 healthcare-09-01631-t005:** Mediating effects of resilience on the relationship between spirituality and self-management.

	Spirituality → Self-management (Model 1)				
	B	SE	β	*t*	*p*
Education level	5.0886	2.527	0.1517	2.0137	0.0459
Monthly income	1.422	1.8349	0.0562	0.775	0.4396
GOLD stage	−0.157	2.1853	−0.0057	−0.0719	0.9428
mMRC grade	−11.0263	4.7641	−0.1805	−2.3144	0.0221
Depression	−7.4397	3.1037	−0.1678	−2.397	0.0178
Spirituality	1.1917	0.215	0.4198	5.5418	<0.0001
	**Spirituality →** **Resilience (Model 2)**				
	**B**	**SE**	**β**	** *t* **	** *p* **
Education level	1.0639	0.752	0.1084	1.4148	0.1593
Monthly income	0.3948	0.546	0.0533	0.7231	0.4708
GOLD stage	1.2054	0.6503	0.1506	1.8536	0.0658
mMRC grade	−4.4063	1.4177	−0.2464	−3.1081	0.0023
Depression	−1.5198	0.9236	−0.1172	−1.6456	0.1020
Spirituality	0.3731	0.064	0.4491	5.8308	<0.0001
	**Spirituality, Resilience →** **Self-management (Model 3)**				
	**B**	**SE**	**β**	** *t* **	** *p* **
Education level	3.7162	2.3578	0.1108	1.5761	0.1172
Monthly income	0.9127	1.7033	0.0361	0.5359	0.5929
GOLD stage	−1.7121	2.049	−0.0626	−0.8356	0.4048
mMRC grade	−5.3419	4.5601	−0.0874	−1.1714	0.2434
Depression	−5.479	2.9028	−0.1236	−1.8875	0.0611
Spirituality	0.7104	0.2215	0.2502	3.2067	0.0017
Resilience	1.29	0.2595	0.3776	4.9715	<0.0001

Note: B = unstandardized regression coefficient; SE = standard error; β = standardized regression coefficient.

**Table 6 healthcare-09-01631-t006:** Mediating model examination by bootstrap.

	Spirituality → Self-Management				
Effect	B	SE	Bootstrap 95% CI	*p*	Effect Ratio
Total effect	1.1917	0.2342	[0.7460–1.6538]	<0.001	/
Direct effect	0.7104	0.2361	[0.2419–1.1722]	0.0017	59.61%
Indirect effect	0.4813	0.1382	[0.2243–0.7671]	/	40.39%

Note: B = unstandardized regression coefficient; SE = standard error; CI = confidence interval.

## Data Availability

The data analyzed in this study are available on reasonable request to the corresponding author.

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
