# Peer review of "Resilience as a Mediator of the Association between Spirituality and Self-Management among Older People with Chronic Obstructive Pulmonary Disease"

_healthcare, 2021, doi:10.3390/healthcare9121631_

Round 1

Reviewer 1 Report

In this study, Chen and Colleagues explored the relationship between spirituality and resilience on self-management among older people with COPD.

The topic of the work is rather interesting and it could be of help to clarify the importance of palliative care in the treatment of chronic diseases. The draft is well written but seems weak in terms of rationale explanation, methodology and data analysis. The rationale of this study was to analyse the impact of spirituality on COPD self-management but either the analyses conducted and the results obtained are unclear. The manuscript should be deeply revised to meet publication standards.

Below are a few items for consideration:

  1. Throughout the manuscript there are some formatting typos, for example, there is no space after the point.
  2. Lines 113-114: Is not clear how many patients were excluded.
  3. Regarding measures and variables, why you didn’t include questions about income, lifestyle (for example smoke), and perceived mental well-being? All those variables could influence self-management more than spirituality and resilience.
  4. Lines 162-163: The correlation analysis should include the other variables indicated in table 1 as potential confounders. For this reason, authors should use multivariate analyses (partial correlation) than univariate analyses.
  5. Lines 178-179: Why have you used mMRC as a dichotomic variable? Are there some references that used this variable in this way?
  6. Table 1 is not clear because the numbers are not aligned. It should be improved.
  7. Lines 173-178: Are there some references that divided the CSMS in three groups? I think is not correct to create the group based on the standard deviation. You should use, for example, cluster analysis to create the three groups.
  8. Table 3: authors should apply partial correlation including demographic variables and the variables related to the degree of COPD and dyspnea as control variables. In addition, I don’t know why the authors didn’t use other variables, such as smoke and perceived mental well-being, to deepen the possible association between spirituality, resilience, and self-management.
  9. Paragraph 3.3: The hypothesized model should be presented in a figure with the predictor variable (x), the outcome (y), and the mediator; in addition, the hypothesized relations should be clarified. Table 4 and Table 5 are not clear. It’s not clear where education level, GOLD stage, and mMCR grade were used as covariates. The covariates should be displayed in the hypothesized model and they should be estimated in the model. The results of the mediation model should be shown in a figure.

The other demographic variables should be used as covariates.

Reviewer 2 Report

It is meaningful to study the relationship between the COPD patient's spirituality and self-management, and examine some variable that mediate the relationship. However, the rationale of the relationship between the variables of the study should be presented. hat's why discussions on the results are inevitably insufficient. In addition, you should have presented only the intended results, and described the results so that the readers can understand them well. When those things are supplemented can they be published in Q1 or Q2 journals. The things that need to be supplemented is as follows.

  1. The first thing that needs to be supplemented is the retentionale for research design. Of course, the most important variable in the study of examining the mediating model is a mediator.

In this study, it is the resilience. However, this manuscript lacks explanation for the resilience. You should have explained the concept of resilience in detail. As you know, the etymology of resilience is 're' + 'silence'. In other words, it means something that calms the mind that was shaken in a stressful or chaotic situation like a frenzy. Therefore, it is more logical when an independent variable or predictor is stress or trauma. Of course, it can be trauma to be diagnosed with a certain diseases such as cancer or coronary heart disease. But it's hard to say that chronic obstructive pulmonary disease doesn't look like that. Therefore, you should somehow highlight in your manuscript that living as a COPD patient is stressful.

  1. If a hypothetical study odel is presented as a picture, it will be easy for the reader to understand the hypotheses and results of the study.

  1. Why did you suggest the difference in research variables according to demographic profile that were not intended? It's better to exclude because it makes it more confusing.

  1. As you presented thr descriptive statistics of each variable, if you presented sub-scale scores of the self-management, it would be better to present it in the correlational matrix.

  1. It is necessary to refer to other papers to write a method of presenting the results of analyzing the mediating effect with Process MACRO.

  1. As I first mentioned about the introduction, the discussion should be emphasized on resilience. Of course, it is important to discuss the significance of spirituality.

Round 2

Reviewer 2 Report

Thank you for modifying what I said that it would be supplemented.

I can see soemethings to correct.

1. The statistical package should be properly named on green. Please change revise macro PROCESS to PROCESS Macro.

2. In Table 3, write the numbers of the self-management subfactors from 3-1 to 3-5.

3. Author contribution: It would be better to describe by the each author.
